# Digital Cytology in Veterinary Education: A Comprehensive Survey of Its Application and Perception among Undergraduate and Postgraduate Students

**DOI:** 10.3390/ani14111561

**Published:** 2024-05-24

**Authors:** Marta Giacomazzo, Francesco Cian, Massimo Castagnaro, Maria Elena Gelain, Federico Bonsembiante

**Affiliations:** 1Department of Comparative Biomedicine and Food Science, University of Padova, 35122 Padova, Italy; marta.giacomazzo@phd.unipd.it (M.G.); massimo.castagnaro@unipd.it (M.C.); 2BattLab, Coventry CV4 7EZ, UK; francesco.cian@hotmail.it; 3Department of Animal Medicine, Production and Health, University of Padova, 35122 Padova, Italy; federico.bonsembiante@unipd.it

**Keywords:** whole slide imaging, telecytology, telepathology, veterinary pathology

## Abstract

**Simple Summary:**

Due to the fast changes in the availability of new technologies, partially accelerated by COVID-19, veterinarians and veterinary students faced the challenges of digital learning technologies embedded in practical laboratories. In particular, veterinary pathology courses transitioned from traditional methods to digital pathology, the viewing of slides on a computer screen. The aims of this study are to evaluate and compare the personal effectiveness and satisfaction, as well as the advantages and disadvantages of digital pathology, specifically digital cytology, as a teaching method among European veterinary students at the undergraduate and postgraduate level who attended digital pathology courses during and before the pandemic. Digital cytology refers to the visualisation of cytological samples on a screen. A Google Form survey consisting of 11 multiple-choice questions was emailed to pathology teachers and distributed to their students. From our results, the main differences emerged in training, the disadvantages of digital cytology, and the preferred method of learning cytology. Finally, all students consider digital cytology as a satisfactory primary teaching methodology; however, the importance of not completely replacing light microscopy, a common tool in everyday veterinary practice, emerged, especially from postgraduate students.

**Abstract:**

The COVID-19 pandemic accelerated technological changes in veterinary education, particularly in clinical pathology and anatomic pathology courses transitioning from traditional methods to digital pathology (DP). This study evaluates the personal effectiveness and satisfaction, as well as the advantages and disadvantages, of DP, in particular digital cytology (DC), as a teaching method among European veterinary students, both at the undergraduate and postgraduate level, who attended digital pathology courses during and before the pandemic. A further aim is to discuss the differences between the two student groups. A Google Form survey consisting of 11 multiple-choice questions was emailed to pathology teachers and distributed to their students. Results indicated that undergraduate students showed greater digital pathology training, favouring DC as the most effective learning modality. In contrast, postgraduate students reported less digital slide training, and their preference for learning cytology was split between DC alone and DC integrated with traditional microscopy. All students experienced whole slide imaging for learning cytology slides prevalently, and they stated that DC enhanced their learning experience. While DC demonstrates personal effectiveness and satisfaction as a teaching method, it is important to not replace pathology training with light microscopy completely, as almost a third of the students indicated.

## 1. Introduction

Veterinary education has undergone a major transformation in recent years in response to rapid technological innovation and subsequent societal demands [1], particularly accelerated by the COVID-19 pandemic. The initial shift during the pandemic involved replacing face-to-face teaching with online content [2] and the adoption of online digital teaching technologies. Particularly, clinical pathology and anatomic pathology courses transitioned from the traditional method, based on the use of glass slides and light microscopes (LMs) [3], to digital slides. This teaching change highlighted the importance of digital pathology (DP) in practical skill development [4]. The term DP, which refers to the viewing of slides on a computer screen, encompasses many technologies and techniques [5]. Essentially, there are three system models for digital pathology currently used: (1) static (store and forward), (2) dynamic (real-time), and (3) whole slide imaging (WSI) [6]. All these models can be applied to evaluate both cytological and histological slides.

Static (‘offline’ or ‘store and forward’): microscope images captured at focal sites using a smartphone or microscope camera are sent for review to a pathologist located at a remote site [7,8].Dynamic (or ‘real-time’): real-time images from a robotic microscope are transmitted to a remote site, allowing the remote operator to control magnification and specimen orientation [9,10].WSI: the entire slide or a selected area of slide (region of interest technique) is digitised using slide scanners or scanning microscopes [8,11,12].

However, limited reports have investigated the effectiveness and satisfaction, as well as the advantages and disadvantages, of DP, especially digital cytology (DC), as a teaching method in the veterinary education setting. The aim of this article is to investigate these aspects among European veterinary students at both the undergraduate and postgraduate levels who attended courses with DP during and before the pandemic. We involved participants from across Europe to expand the sample size, aiming to capture not only the perceptions of Italian students or students from a specific state. An additional aim is to discuss and compare different perceptions between undergraduate and postgraduate European students, another aspect where, to our knowledge, the literature is limited.

## 2. Materials and Methods

A Google Form containing an anonymous online survey was sent via e-mail to European undergraduate as well as postgraduate veterinary students that attended online courses or webinars based on DP from January 2021 to June 2021. For postgraduate students only, the survey link was also shared on the European College of Veterinary Clinical Pathology (ECVCP) and European Society of Veterinary Clinical Pathology (ESVCP) and European College of Veterinary Pathologists (ECVP) websites and through the Zoom chat at the end of online webinars.

The survey consisted of 11 multiple-choice questions (refer to Appendix A). The authors developed the questionnaire based on their experience teaching digital cytology and drawing from the questions and discussions that emerged over years of interaction with students. The questions covered demographic information of participants such as age, year of graduation, current position (e.g., PhD student, post-doc fellow, EBVS resident, general practitioner), or years of attendance and at which university. Then, the survey investigated the courses attended for both digital and traditional pathology, the types of digital slides presented during the courses (histological, cytological, and haematological slides), and the modality of digital pathology (DP) experienced. Finally, the survey focused on digital cytology (DC), asking the preferred modality for learning DC, the associated pros and cons (with the possibility of choosing multiple answers without the need to place them in order of importance), and whether DC improved the cytology skills. The last question explored a comparison between traditional and DC as the preferred cytology learning methodology, with an opportunity for respondents to provide comments.

A statistical analysis was conducted using Pearson’s chi-squared test and z-test via RStudio Software (http://www.rstudio.com/ (accessed on 3 May 2024)) to determine if a statistically significant difference between the answers provided by PGSs and UGSs existed. The analysis focused on questions of utmost relevance to our study, namely, the advantages and disadvantages of DC, the preferred modality of learning DC, and whether DC improved cytology skills.

## 3. Results

### 3.1. Undergraduate Students (UGSs)

A total of 176 undergraduate students participated in the survey, with the majority attending the third year of the veterinary medicine course (35%), followed by students attending the first year (19%), the fifth year (13%), and the second and fourth years (8% and 7%, respectively). Eighteen percent of the students did not provide an answer.

Most of the UGSs attended Italian (48%) and French (39%) universities while only 3% attended German universities. Ten percent of UGSs did not respond (see Figure 1).

Regarding the age distribution, 90% of the students were between 19 and 25 years old, 8% between 26 and 35 years old, and 1% between 36 and 50 years old; 1% did not provide an answer. Based on the age group most represented, we could consider the UGSs as Generation Z.

Almost half of the UGSs (49%) attended up to three courses with glass slides, 18% attended more than five courses, 18% did not attend any courses, and 14% attended up to five courses. Only 1% of the students did not provide an answer. In contrast, for DP courses, 51% of students attended more than five courses, 38% attended up to three courses, 7% attended up to five courses, and 4% did not attend any courses (see Table 1).

During DP courses, the majority of UGSs were exposed to histological, cytological, and haematological slides (45%) (see Table 2).

Among the digital cytology courses, the most used modality to show the digital slides was the WSI (35%) (see Table 3).

WSI was indicated as the favoured modality to learn cytology by 44% of UGSs. The second was a robotic optical microscope with a camera remotely controlled by the teacher (16%), followed by glass cytology slides (15%) and picture-based cases (8%). However, 15% of the students used only one of these methods and were therefore unable to compare and express a preference. Two percent of the students did not respond (Table 4).

The main advantage of learning with DC, as reported by 78% of the UGSs, is the possibility to access digital cases at any time. Other important advantages indicated included freedom to navigate throughout the digital slide (63%), the limited equipment needed (48%), and the absence of time limits (30%). Additionally, the advantages listed in the survey included the certainty of everyone viewing the same pictures (3.5%), the possibility to discuss together what you are seeing (1%), and the absence of headaches induced by optical microscopes (1%). Less than 4% of the UGSs did not provide an answer.

The principal disadvantage of DC, as reported by 40% of the UGSs, is the technical aspect. Other disadvantages included inadequate resolution and/or different colours compared to glass slides or microscopic pictures (27%), the lack of guidance when navigating digital slides (26%), and the longer learning curve required to adapt to digital slide assessment (12%). Eighteen percent of the students did not answer.

The results of the advantages and disadvantages are listed in Table 5.

The majority of students (93%) believe that DC could improve cytology skills, with only 3% of the students holding the opposite view, while 4% of the students did not respond.

When comparing learning with DC and with traditional cytology courses, 43% of the UGSs indicated that DC is better than traditional cytology for learning purposes, while 34% of the UGSs reported that DC is well accepted as integration. Fourteen percent (14%) of the UGSs reported that they had never attended any traditional cytology course. Only 4% of the students preferred traditional cytology courses, and 6% did not provide a response).

Three comments were submitted at the end of the survey, emphasising the importance of learning and the use of light microscopes.

### 3.2. Postgraduate Students (PGSs)

Sixty-eight (*n* = 68) PGSs participated in the survey. The majority of the PGSs worked as general practitioners (78%) followed by PhD students and post-doc fellows (9%), residents (five clinical or anatomic pathology residents and one in another discipline for a total of 9%), and two clinical pathologists (3%); one PGS (<1%) did not answer.

Only fifty percent of the PGSs specified the university of veterinary medicine they previously attended. The universities attended were distributed in 11 European countries and the majority of PGSs (44%) attended Italian universities (see Figure 2).

Sixty-two percent (62%) of the veterinarians were between 26 and 35 years old, 23% between 36 and 50 years old, 12% over 50 years old, and 3% between 19 and 25 years old. Based on the age group most represented, we could consider the PGSs as Generation Y.

Almost half of the PGSs (47%) attended one to three courses with glass slides, 25% did not attend any courses, 16% attended more than five courses, and 12% attended four to five courses. In contrast, for courses with digital slides, 79% of the PGSs attended one to three courses, 16% did not attend any courses, 3% attended up to five courses, and 2% attended four to five courses (refer to Table 1).

During DP courses, the majority of postgraduate students were exposed to only cytological slides (74%) (see Table 2).

DC slides were mostly evaluated using WSI (52%), followed by a mixture of WSI, static, and dynamic techniques (see Table 3).

Forty-four percent of the PGSs favoured WSI as their preferred method to learn cytology, closely followed by glass cytology slides (40%). Only 4% of the PGSs preferred dynamic DC, while 3% indicated static DC. However, 9% of the PGSs used only one of these methods and were therefore unable to compare and express a preference (see Table 4).

The primary advantage of DC learning, as indicated by 82% of the PGSs, is the possibility of accessing digital cases at any time. Other advantages mentioned include the freedom of navigating throughout the digital slide (56% of the PGSs), the limited equipment needed (46%), and the absence of time limits (32%). A minority of students also cited additional benefits, such as the certainty of everyone viewing the same frame (1%), the possibility to discuss together what all see (1%), and the possibility of storing cytological slides indefinitely (1%). Almost 4% of the PGSs did not respond.

The principal disadvantage of DC, as reported by 60% of the PGSs, is the inadequate resolution and/or different colours compared to glass slides or microscopic pictures. Other notable disadvantages mentioned include the technical problems related to the software used (21% of responses), the lack of guidance when navigating digital slides (13%), and the longer learning curve required to adapt to digital slide assessment (12%). Twelve percent (12%) of the PGSs did not answer.

The advantages and disadvantages of DC are listed in Table 5.

The majority (93%) of the PGSs believe that DC could improve cytology skills, with only 6% of the PGSs believing the contrary. Only 1% of the PGSs did not respond.

When comparing learning preferences between DC and traditional cytology courses, the PGSs were almost evenly divided: about 32% expressed a preference for traditional cytology courses; however, they also indicated a positive acceptance of integration with DC. Almost 30% of the PGSs believed that digital cytology was better than traditional cytology for learning purposes. A total of 13% of the PGSs preferred traditional cytology courses without the use of DC. Twenty-four percent (24%) of the students reported having never attended any traditional cytology courses, while 1% did not provide a response (Figure 3).

The survey received five comments that expressed a strong preference for learning with DC; however, three comments highlighted resolution and connection problems.

### 3.3. Comparison between the UGSs and PGSs

To summarise the main differences, Table 6 compares the predominant responses given by both groups of students for each question.

No statistically significant difference emerged on the question regarding the advantages of DC between the responses provided by the PGSs and the UGSs (X-squared = 4.71, *p*-value = 0.789). Instead, the question concerning the disadvantages of DC revealed a statistically significant difference (X-squared = 23.73; *p*-value < 0.001), specifically in the technical problems (Z-test = 5.67, *p*-value = 0.017), predominantly acknowledged by the UGSs (40%), and in the inadequate resolution (Z-test = 22.40, *p*-value < 0.001), predominantly affirmed by the PGSs (60%).

Moreover, the question concerning the preferred way to learn cytology revealed a statistically significant difference between the responses provided by the PGSs and UGSs (X-squared = 22.39, *p*-value = 0.0001676). In particular, this difference was observed in a robotic optical microscope with a camera remotely controlled by a teacher (Z-test = 5.85, *p*-value = 0.01561), a method favoured by 16% of the UGSs, and in the glass cytology slides (Z-test = 16.90, *p*-value < 0.001), favoured by 40% of the PGSs.

Finally, a statistically significant difference between the responses provided by the PGSs and UGSs was observed in the modality in which the student learned more (X2 = 9.41; *p*-value = 0.024), particularly in the response related to the preference for traditional cytology courses without the use of DC (Z-test = 6.86; *p*-value = 0.008), as reported by 13% of the PGSs (Figure 3).

## 4. Discussion

The current study aimed to assess the perceptions and preferences between the digital and traditional cytology learning methods prior to and during the pandemic among two different groups, UGSs and PGSs.

In terms of demographics, the results showed that the majority of the UGSs respondents, mainly from Italy and France, were between 19 and 25 years old and in their third year of veterinary studies. On the other hand, the PGSs, predominantly aged between 26 and 35 years and mainly from Belgium or Italy, were mostly employed as general practitioners. Unfortunately, data on the country of origin were too widespread throughout Europe for the PSGs and restricted to France and Italy for the UGSs to draw some conclusions on the possible effects of this variable.

The majority of participants from both groups attended 1–3 courses with traditional slides. However, one notable difference between the two groups was their attendance in courses with digital slides. In fact, the UGSs attended more courses with digital slides (more than five courses attended) than the PGSs (1–3 courses attended) and were more exposed to different types of digital slides, namely histological, cytological, and haematological slides. In contrast, the PGSs were predominantly exposed to only DC slides.

Specifically, the WSI was the most utilised and preferred DC learning methodology for both groups. In veterinary medicine, WSI is a powerful pedagogical tool: it improves cytology practical assessment [12] and facilitates histology teaching, learning, and assessment [13,14,15] compared to LMs. WSI offers the ability to learn collaboratively, a more effective use of time, and the flexibility of online learning [15]. Generally, student perceptions of the VM system are generally very positive [16]. However, to our knowledge, the existing literature predominantly contrasts DP (or DC) education with traditional microscopy education, while a comparative analysis of various DP methodologies regarding learning preferences, advantages, and disadvantages remains largely unexplored.

Although the students were not asked the reason for this preference, we can assume that, in our study, the UGSs and PGSs favoured WSI due to its familiarity, being the method they used most frequently. Additionally, WSI resembles the daily clinical routine of LMs, offering the advantage of evaluating entire slides rather than selected microscopical fields, unlike static DC, and providing independence in assessment, unlike robotic DC.

A statistically significant difference emerged between two answers: the robotic optical microscope with a camera remotely controlled by the teacher, reported by 16% of the UGSs as opposed to 4% of the PGSs, and the glass cytology slides, favoured by 40% of the PGSs compared to 15% of the UGSs.

The reported advantages of DC courses were similar between the two groups, with no statistically significant differences emerging. The primary benefit reported was the ability to assess cases at any time with limited equipment: DC allows access to online educational files on multiple device types, without time restriction, in a variety of locations [17] and simultaneously [17,18]. These aspects imply the possibility of collaborative learning [4,19], enabling students to compare digital slides and discuss features with instructors [16] and peers.

However, the majority of the PGSs reported the inadequate resolution of DC, while the UGSs cited technical problems, with a statistically significant difference. The image quality of digitalised slides exposed in the various courses remains unexplored. In the literature, the limitation linked to image quality is well documented [16], even if contrasting [19]. Part of the reason for the limitation may be related to the specimens used [16], the scanner (as reported in our study), the programmes used, and the connection. In fact, the most common problems associated with online education in general included the availability of internet and electronic devices, the transmission speed, which is often insufficient [12,20,21,22], and the cost of the internet [21].

These findings suggest that digital preparation is required [23], emphasising the need to improve the online veterinary education infrastructure as much as possible [21], including the provision of adequate server support, internet speeds, and other appropriate measures [12].

Moreover, the majority of the UGSs preferred learning cytology through digital slides or integrating the digital approach with traditional methods. Instead, the PGSs exhibited a more divided learning preference, with some favouring the DC and some a combination of DC with the LMs. However, if only 4% of the UGSs indicated the traditional learning method as the best, 13% of the PGSs preferred it, with a statistically significant difference.

The latest observed differences—particularly in disadvantages and learning preferences—could be attributed to varying backgrounds, namely experiences—social and professional—and training, within the two groups. In fact, the technological progress observed in recent years has increased the digital divide according to age [24], and not only in education. In our study, we could divide the UGSs and the PGSs into Generation Z (Gen Z) and Generation Y (Gen Y), respectively. Gen Y and Gen Z reveal significant differences across various aspects, such as teaching preferences, learning styles, technology usage, and communication methods [25].

Gen Y, also known as Millennials, born between 1980 and 1995, grew up during the rapid expansion of the internet and digital technology [25]. Consequently, they are often referred to as “digital natives” [26]. Millennials prefer interactive, self-paced, technology-based methods in education, and their learning style is characterised by collaboration and networking [25] or blended learning [26]. So, they experience both traditional and innovative learning methods [ed.]. Despite the fact that the majority of all students attended 1–3 courses with traditional slides, it is crucial to specify that most of the PGSs had more experience with them. In fact, as the majority of the PGSs were general practitioners, they routinely used LMs in their clinical practices [16]—and not the digital microscope—for evaluating cytological specimens [16]. Moreover, during their veterinary degree courses, they were probably exposed only to LMs. As a result, they were more likely to recognise the differences in image quality between digital and traditional slides, while acknowledging the advantages of DC. Therefore, to achieve a balance between experience and e-learning, they are more inclined to complement DC with LMs for training purposes.

In contrast, the UGSs were more exposed to and trained in the digital world, possibly influenced by the increasing prevalence of e-learning in veterinary education [27]. They belong to Gen Z, born between 1995 and 2012, and are the first generation to grow up with constant access to digital technology and social media [25]. They rely on the internet as a primary general information source [28] and prefer hybrid learning approaches that incorporate technology and multimedia content instead of text [29] or traditional frontal lecture [30]. Moreover, Gen Z tends to favour individual learning environments [29], and in our study, it is plausible to assume that the UGSs may lack experience with LMs [14], with LMs being used for limited periods and primarily for learning purposes. For all these reasons, it can be assumed that they prefer DC over traditional cytology, recognising technical aspects as disadvantages. Despite that, one-third of the UGSs feel the necessity of learning with LMs, and DC is well accepted as integration. This is coherent with the literature. Adequate time and training in the curriculum should be dedicated to developing LM skills for the routine evaluation of cytological specimens [3,31]. As yet reported, the UGSs considered the use of LMs and associated skills important and necessary. They felt that LMs should not be completely replaced [12,16,32] or could be irreplaceable [14], requiring more or additional training [3,33] for fear that they would not acquire sufficient LM skills [20].

Finally, all students (UGSs and PGSs) unanimously agreed that DC could improve the cytology learning process, indicating veterinary students’ appreciation for teaching with DP, as previously reported [4,12,16,18].

One major limitation of this study is the uneven sample sizes in the compared populations of students, namely UGSs and PGSs (n = 176 and n = 68, respectively), and their relative different backgrounds and experiences. Additionally, the potential bias in subjective responses needs consideration, as there were variations in how students answered certain questions. This bias is also associated with the questionnaire format; specifically, students who never attended courses with digital slides were allowed to answer all survey questions equally. However, these students represented approximately 4%, therefore having little impact on the overall result. Moreover, the didactic objectives and details of courses attended were not investigated. Finally, all participants were European, necessitating further research to assess the applicability of the DC learning methodologies in different countries and thus compare any potential differences in terms of both perceptions and performance.

## 5. Conclusions

In conclusion, our findings indicate similarities in the results obtained from both the UGSs and PGSs, with key differences primarily attributed to background, training, and preferences for learning cytology. The UGSs showed greater training in digital pathology, encompassing various slide preparations. Most of them identified DC as the most effective personal modality for learning, possibly influenced by the increase in veterinary e-learning, while highlighting the technological aspect as a limit. Conversely, the PGSs reported less training in digital slide courses and were divided on their preference for the way of learning cytology: DC versus LMs in conjunction with DC, while a portion remained committed solely to LMs. This aspect may be influenced by their daily routine, background, and experience, with image quality being reported as a major limit. However, the PGSs reported the same DC advantages described by the UGSs. All students experienced prevalently WSI for leaning cytology slides and reported that DC improved their cytology leaning. Finally, our results highlight the effectiveness and satisfaction of DC as the primary teaching method, and that it is important to integrate and not completely replace light microscopy as it remains a common tool in daily veterinary practice.

## Figures and Tables

**Figure 1 animals-14-01561-f001:**
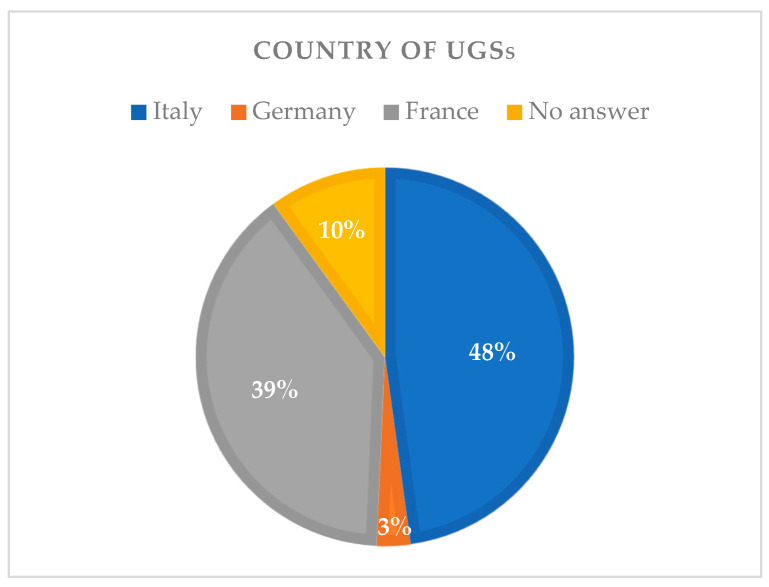
European countries of origin of the universities attended by undergraduate students (UGSs).

**Figure 2 animals-14-01561-f002:**
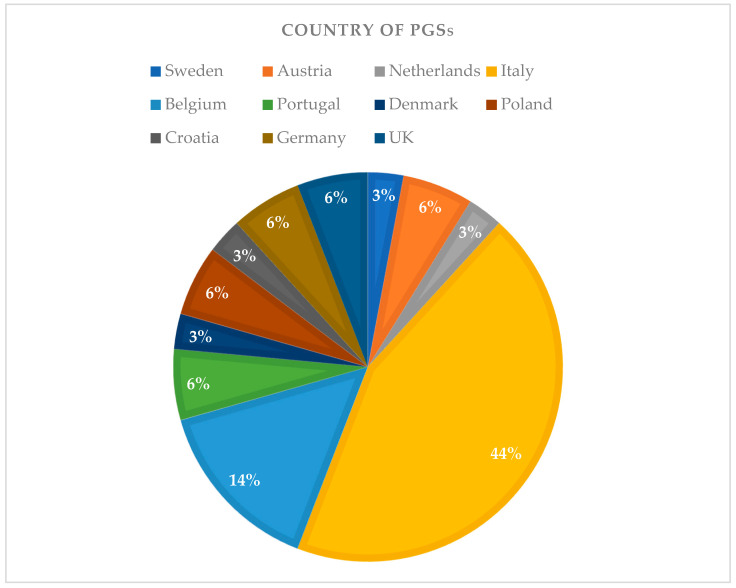
European countries of origin of the universities attended by postgraduate students (PGSs).

**Figure 3 animals-14-01561-f003:**
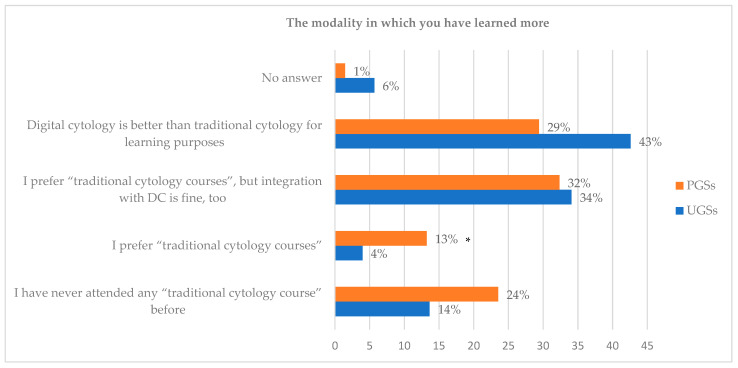
The modality in which students have learned more according to their opinion (UGSs = undergraduate students; PGSs = postgraduate students; DC = digital cytology). PGSs preferred traditional cytology courses without the use of DC (* *p*-value = 0.008).

**Table 1 animals-14-01561-t001:** Attendance of courses with cytological digital slides and with glass slides (UGSs = undergraduate students; PGSs = postgraduate students).

Attendance of Courses	With Glass Slides (UGSs)	With Glass Slides (PGSs)	With Digital Slides (UGSs)	With Digital Slides (PGSs)
>5	18%	3%	**51%**	16%
4–5	14%	2%	7%	12%
1–3	**49%**	**79%**	38%	**47%**
0	18%	16%	4%	25%
No answer	1%	0%	0%	0%

The bold underlines the prevailing response.

**Table 2 animals-14-01561-t002:** Types of digital slides used during digital cytology courses (HiDSs = histological digital slides; CDSs = cytological digital slides; HeDSs = haematological digital slides; UGSs = undergraduate students; PGSs = postgraduate students).

Types of Digital Slides	Percentage of Answers (UGSs):	Percentage of Answers (PGSs):
HiDSs, CDSs, and HeDSs	**45%**	4%
HiDSs and CDSs:	19%	9%
HiDSs and HeDSs	14%	0%
CDSs	10%	**74%**
HiDSs	8%	7%
CDSs and HeDSs	3%	3%
HeDSs	1%	0%
No answer	0%	3%

The bold underlines the prevailing response.

**Table 3 animals-14-01561-t003:** Modality of digital cytology case presentation during the courses (WSI = whole slide imaging; DC = digital slide; UGSs = undergraduate students; PGSs = postgraduate students).

Modality	Percentage of Answers (UGSs):	Percentage of Answers (PGSs):
WSI	**35%**	**51%**
Pictures of selected areas of a cytology glass slide and WSI (static DC and WSI)	21%	15%
Pictures of selected areas of a cytology glass slide and robotic optical microscope with a camera remotely controlled by the teacher and WSI (static DC, dynamic DC, and WSI)	16%	9%
Pictures of selected areas of cytology glass slides (static DC)	11%	12%
Robotic optical microscope with a camera remotely controlled by the teacher (dynamic DC)	6%	2%
Robotic optical microscope with a camera remotely controlled by the teacher and WSI (dynamic DC and WSI)	5%	4%
Pictures of selected areas of cytology glass slides and a robotic optical microscope with a camera remotely controlled by the teacher (static DC and dynamic DC)	3%	4%
No answer	3%	3%

The bold underlines the prevailing response.

**Table 4 animals-14-01561-t004:** The favourite way to learn cytology indicated by students (WSI = whole slide imagining; UGSs = undergraduate students; PGSs = postgraduate students).

The Favourite Way to Learn Cytology	Percentage of Answers (UGSs)	Percentage of Answers (PGSs)
WSI	**44%**	**44%**
Robotic optical microscope with a camera remotely controlled by a teacher	16%	4%
Glass cytology slides	15%	40%
I used only one of these methods, so I am unable to compare and express a preference	15%	9%
Picture-based cases	8%	3%
No answer	2%	0%

The bold underlines the prevailing response.

**Table 5 animals-14-01561-t005:** Advantages and disadvantages of digital cytology courses (PROS = advantages; CONS = disadvantages; UGSs = undergraduate students; PGSs = postgraduate students).

PROS	UGSs	PGSs	CONS	UGSs	PGSs
Possible to access digital cases when I want/need	**78%**	**82%**	Technical problems with using the dedicated software to visualise digital slides	**40%**	21%
Freedom of navigation throughout the digital slide	63%	56%	Inadequate resolution and/or different colours compared with glass slides or microscopic pictures	27%	**60%**
Limited equipment needed (no microscope)	48%	46%	Lack of guidance when navigating throughout the digital slide	26%	13%
No time limit	30%	32%	Longer learning curve to adjust to digital slide assessment	12%	12%
You are sure that you are looking at the same structures as your teacher	3.5%	1%	No answer	18%	12%
Possibility to discuss together what we see	1%	1%			
No microscope-induced headache	1%	0%			
Possibility to keep slides forever	0%	1%			
No answer	3.5%	4%			

The bold underlines the prevailing response.

**Table 6 animals-14-01561-t006:** The predominant responses given by both groups of students for each question (WSI = whole slide imaging; DC = digital cytology; UGSs = undergraduate students; PGSs = postgraduate students).

	UGSs (*n* = 176)	PGSs (*n* = 68)
Age	19–25 years (90%)	26–35 (62%)
Most attended courses with glass slides	1–3 (49%)	1–3 (79%)
Most attended courses with digital slides	>5 (51%)	1–3 (47%)
Types of digital slides used in the courses	Histological, cytological, and haematological slides (45%)	Cytological slides (74%)
Modality of cytological digital cases’ presentation	WSI (35%)	WSI (51%)
The favourite way to learn cytology	WSI (44%)	WSI (44%)
Main advantages of using DC	Possibility to access digital cases when I want/need (78%); freedom of navigating throughout the digital slide (63%); limited equipment needed (no microscope) (48%); no time limit (30%)	Possibility to access digital cases when I want/need (82%); freedom of navigating throughout the digital slide (56%); limited equipment needed (no microscope) (46%); no time limit (32%)
Main disadvantages of using DC	Technical problems with software (40%); inadequate resolution (27%); lack of guidance when navigating (26%)	Inadequate resolution (60%); technical problems (21%)
Did the training with DC help you to improve your skill?	Yes (93%)	Yes (93%)
The modality in which you learned more	DC is better than traditional (43%)/preference for traditional cytology course, but integration with digital cytology is fine (34%)	Preference for traditional cytology course, but integration with DC is fine (32%)/digital pathology is better (29%)

## Data Availability

Data are unavailable due to privacy or ethical restrictions.

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
