# Peer review of "Digital Cytology in Veterinary Education: A Comprehensive Survey of Its Application and Perception among Undergraduate and Postgraduate Students"

_animals, 2024, doi:10.3390/ani14111561_

Round 1

Reviewer 1 Report

Comments and Suggestions for Authors

Thank you for the opportunity to read this paper. In general it is a good paper, solid research, and interesting research questions. I have a few comments, which I believe you should address. 

1.    It is some unclarity about the questionnaire. I would like for you to write a little bite more about it in the method section. What I’m mostly concerned about is the options in response, e.g. (in table S1) question 8 and 9 – which are similar have different response options. It can be important to know whether the respondents can answer with more than one or can list in certain order etc.. Also, another issue, in line 156 and 205 you talk about comments from the questionnaire, but in the method section you have not explained that comments were a possibility for the responders. I would like you to elaborate on the design of the questionnaire, including reasons for the choices you have made.  

2.    I don’t think you get enough out of the fact that you have responders from different nations and cultures. You talk a lot about age and generations, but the location is not pursued. I think you need to address this. Why asking for respondents across Europe? And to what extent was that why answered in the survey data. Probably you cannot say anything or very little, but then that should be made clear, since you have set up the design as you have.

3.    Line 88 says: “we could consider UPG as”. I do not know what UPG stands for – should perhaps be UGS?

4.    Line 278 says: “Gen Y (or Millennials) and”. I suggest removing the ‘(or Millennials)’, since you in line 281 says the same thing, but much more elegantly: “Gen Y, also known as Millennials, born”.

5.    Line 295 says: “trained to digital word, p”. I do not understand why it says ‘word’ – should it say ‘world’?

Author Response

  1. It is some unclarity about the questionnaire. I would like for you to write a little bite more about it in the method section. What I’m mostly concerned about is the options in response, e.g. (in table S1) question 8 and 9 – which are similar have different response options. It can be important to know whether the respondents can answer with more than one or can list in certain order etc. Also, another issue, in line 156 and 205 you talk about comments from the questionnaire, but in the method section you have not explained that comments were a possibility for the responders. I would like you to elaborate on the design of the questionnaire, including reasons for the choices you have made.

A: We forgot to specify that, for both question 8 and 9 there was the possibility to choose more option. We have corrected question 8 posed in table S1 and we stated that pros and cons questions allowed the possibility of choosing several answers without the need to place them in order of importance in the materials and methods section from line 91 to line 92 and in line 94 explained the possibility to add comments.

The questionnaire and its questions were developed by the authors of the article based on their experience teaching digital cytology and drawing from the questions and discussions that emerged over years of interaction with students and other teachers in veterinary pathology and cytology (as added from line 81 to line 84). In particular, the teaching situation during and the pandemic arise some concern about the new technology in teaching pathology could be faced by students with different age and experience. 

  1. I don’t think you get enough out of the fact that you have responders from different nations and cultures. You talk a lot about age and generations, but the location is not pursued. I think you need to address this. Why asking for respondents across Europe? And to what extent was that why answered in the survey data.Probably you cannot say anything or very little, but then that should be made clear, since you have set up the design as you have.

A: We agree with the reviewer that also the country of origin of the responder could be an influencing factor. Unfortunately, only 50% of the PGS had answered this question and the other 50% are spread across Europe making difficult to drawn a conclusion. As regards UGS, the majority of the responder (87%) were from Italy and France thus it seems restrictive to us comparing only two countries.

We added the University attended from line 189 to line 199, line 280, from line 281 to line 282, from line 282 to line 285. These limits were reported from line 385 to line 386.

We requested participants from across Europe to expand the sample size, aiming to capture not only the perceptions of Italian students or students from a specific state even if the comparison between different populations was not our aim. (These sentences have been added from line 69 to 71).

  1. Line 88 says: “we could consider UPG as”. I do not know what UPG stands for – should perhaps be UGS?

A: Yes, we corrected the mistake.

  1. Line 278 says: “Gen Y (or Millennials) and”. I suggest removing the ‘(or Millennials)’, since you in line 281 says the same thing, but much more elegantly: “Gen Y, also known as Millennials, born”.

A: We corrected the line.

  1. Line 295 says: “trained to digital word, p”. I do not understand why it says ‘word’ – should it say ‘world’? Yes, we corrected the mistake.

Reviewer 2 Report

Comments and Suggestions for Authors

This is a timely and interesting topic that is well-written. Unfortunately, there is no true research methodology. Though it is stated that distance education teaching techniques are helpful but should not wholly replace in-person education, the lack of statistical analysis does not provide adequate support for these conclusions. Gross percentages can be very useful in making suggestions, but without some accompanying descriptive statistics, the conclusions are not as strong as they could be. The alternative would be to employ a qualitative approach and request focus group/interviews with participants, but I suspect this would be much more difficult to complete given the nature of the survey. Strongly recommend the employment of descriptive statistical analysis to provide more robust support to your conclusions, as otherwise this ends up being a report of findings and may not have the wider audience it deserves.

Comments on the Quality of English Language

The English writing was excellent. There is one possible typo on line 295 - is "word" supposed to be "work"? And a word used in line 318 - "provenience" - which appears to be out of context. Perhaps the authors intended to indicate "origins"? Whether or not this is the case, could leave this word out as "background and experiences" likely conveys what is intended.

Author Response

  1. This is a timely and interesting topic that is well-written. Unfortunately, there is no true research methodology. Though it is stated that distance education teaching techniques are helpful but should not wholly replace in-person education, the lack of statistical analysis does not provide adequate support for these conclusions. Gross percentages can be very useful in making suggestions, but without some accompanying descriptive statistics, the conclusions are not as strong as they could be. The alternative would be to employ a qualitative approach and request focus group/interviews with participants, but I suspect this would be much more difficult to complete given the nature of the survey. Strongly recommend the employment of descriptive statistical analysis to provide more robust support to your conclusions, as otherwise this ends up being a report of findings and may not have the wider audience it deserves.

A: Thank you for the suggestion that could help improve our manuscript. We conducted statistical analyses that revealed some statistically significant differences between the groups. We added the tests and software used from line 96 to line 100. Moreover, we added the statistical results from line 252 to 269, from line 307 to line 310, line 313, line 321, line 339 and Figure 1 (added later, from line 270 to line 271).

  1. There is one possible typo on line 295 - is "word" supposed to be "work"? And a word used in line 318 - "provenience" - which appears to be out of context. Perhaps the authors intended to indicate "origins"? Whether or not this is the case, could leave this word out as "background and experiences" likely conveys what is intended.

A: We corrected the mistake and left background and experiences

Reviewer 3 Report

Comments and Suggestions for Authors

I am a bit confused about the educational problem the study is intended to address. There seem to be several lines of inquiry: effectiveness of the use of technology that arose as a result of the pandemic; a comparison of undergraduate students' and postgraduates' preferences for different applications of digital images for learning cytology and histopathology (which may be related to when they were born-- Generation Y or Z); and general questions of effectiveness, preference and learner satisfaction with the use of digital images to learn in this discipline.

Perhaps the use of the word "Impact" in the title suggested to me that the focus of the study would be on the effectiveness of digital images to support learning, as measured by exams or other artifacts of students' work. (What does good learning in cytology look like, and how do these various image formats support that? Are there signifcant differences among the image types that can help or hinder learning?) Instead, the study seemed primarily focused on satisfaction with and preference for different types of digital images, comparing the UGS and PGS populations. The authors may consider whether a revision in the title might more accurately describe the study.

As I read the manuscript, I found myself wondering how the different types of images facilitated students' achievement of the instructor's educational objectives. Are some more effective than others? If so, why, and if not, does it matter which image format is used? It's possible that learners' familiarity with some image types influences their preferences. But the question that remains unanswered here seems to be whether students are learning better, or as well, with different types of image formats, and not necessarliy what generation they belong to (Y or Z).

Student data in this study came from six institutions. Are the student populations equivalent? A little more background would help support the generalizability of the data. Similarly, it would be helpful to know more about the courses in which the digital images are used (do they have the same educational objectives, for example? Are the images of the same quality across institutions?) 

Data were collected using an 11-item survey, nearly half of which focused on demographics. It would be helpful if there was more detail about the courses students would have already completed, to know whether there is a relationship between familiarity (having seen and practiced with certain kinds of images prior to the study) and students' preferences. Also, it dd not appear as though the differences between the two student populations (UGS and PGS) were statistically significant. Was this expected? Does it matter? 

In general, self-reported data is not reliable, so it would be helpful to include data from student performance demonstrating students' success using the images in different formats. This would be particularly useful for instructors who wonder whether they should make changes in their instructional approaches as a result of the  study.

Other comments: the references cited are very appropriate and provide a good foundation for the study. The tables/supplemental materials are very clear. They are easy to interpret and support the authors' conclusions.

Author Response

I am a bit confused about the educational problem the study is intended to address. There seem to be several lines of inquiry: effectiveness of the use of technology that arose as a result of the pandemic; a comparison of undergraduate students' and postgraduates' preferences for different applications of digital images for learning cytology and histopathology (which may be related to when they were born-- Generation Y or Z); and general questions of effectiveness, preference and learner satisfaction with the use of digital images to learn in this discipline. Perhaps the use of the word "Impact" in the title suggested to me that the focus of the study would be on the effectiveness of digital images to support learning, as measured by exams or other artifacts of students' work. (What does good learning in cytology look like, and how do these various image formats support that? Are there significant differences among the image types that can help or hinder learning?) Instead, the study seemed primarily focused on satisfaction with and preference for different types of digital images, comparing the UGS and PGS populations. The authors may consider whether a revision in the title might more accurately describe the study.

  • Reviews comments are correct and the lines of inquiry are several. Overall, our main objective was to study students' perceptions of their preference for cytology learning. Therefore, we change the title accordingly by replacing “impact” with “perceptions”.
  1. As I read the manuscript, I found myself wondering how the different types of images facilitated students' achievement of the instructor's educational objectives. Are some more effective than others? If so, why, and if not, does it matter which image format is used? It's possible that learners' familiarity with some image types influences their preferences. But the question that remains unanswered here seems to be whether students are learning better, or as well, with different types of image formats, and not necessarily what generation they belong to (Y or Z).

A: We believe that different types of images are more effective in learning, but this effect depends on the learner. It is undoubtedly true that the familiarity of the student with a type of image can make them prefer it and thus facilitate the learning process. We also believe that different generations have been exposed to different types of images in day-to-day life and during the different classes of school and that these two variables (familiarity with certain images and generations) are closely linked. For these reasons we cited generations in order to justify the differences observed - particularly in disadvantages and learning preferences, as reported from line 341.

It’s correct that we did not evaluate the efficacy in learning. The purpose of our study was not to compare, in terms of objective evaluation, such as scoring, the different cytological digital learning methodologies, but only to assess the self-perception of students.

The answer is reported from line 298 to line 301 - and we added the word ‘learning’ - and from 391 to line 395. The image format is important for resolution, as reported from line 322 to line 328.

  1. Student data in this study came from six institutions. Are the student populations equivalent? A little more background would help support the generalizability of the data. Similarly, it would be helpful to know more about the courses in which the digital images are used (do they have the same educational objectives, for example? Are the images of the same quality across institutions?) 

A:  In the two clusters of students (UGS and PGS), students from the population vary depending on the year of graduation (UGS) and the year of experience (PGS), as well as differences related to the different countries of origin. We asked participants from all over Europe to expand the sample size in order not only to capture the perceptions of Italian students or students from a particular country. However, the comparison between different populations was not our objective (These sentences have been added from line 69 to 71).

We added the University attended by participants from line 189 to 199, line 280, from line 281 to line 282 and from line 282 and line 285. The limitation of our study related to the heterogeneity of the population was reported from line 384 to line 386.

 With regard to the teaching objectives of the courses, we did not include any question in our survey. However, it is reasonable that the aim of the cytology course for undergraduate students was to teach the basic approach to diagnosis of cytology. In the case of the PSG, the majority of respondents had attended one to three courses, so it is reasonable that these were basic courses in diagnostic cytology, but we do not have specific information on this subject.  We added this limit from line 391 to line 392.

We also have no information about the image’s quality of the courses, as we added from line 321 to line 322. As far as image quality is concerned, we only know that it is not an advantage of digital cytology, as reported from line 320 to line 321, in Table 6 and Table 5, from line 160 to line 164 and from line 226 to line 230 and from line 259 to line 264 (added later). Moreover, we added these aspects from line 391 to line 392.

  1. Data were collected using an 11-item survey, nearly half of which focused on demographics. It would be helpful if there was more detail about the courses students would have already completed, to know whether there is a relationship between familiarity (having seen and practiced with certain kinds of images prior to the study) and students' preferences. Also, it dd not appear as though the differences between the two student populations (UGS and PGS) were statistically significant. Was this expected? Does it matter?

A: The details of courses remains unexplored so we are not able to answer, as we have added from line 391 to line 392. About preferences and familiarity, we discussed from line 302 to line 304 and from line 341 to 379.

We add a statistical analysis: we added the tests and software used from line 96 to line 100, and the results from line 252 to 269, from line 307 to line 310, line 313, line 321, line 339 and Figure 1 (added later, from line 270 to line 271).

  1. In general, self-reported data is not reliable, so it would be helpful to include data from student performance demonstrating students' success using the images in different formats. This would be particularly useful for instructors who wonder whether they should make changes in their instructional approaches as a result of the study.

A: Although it is fundamental to evaluate the learning effectiveness of digital cytology compared to traditional slides, this is not our purpose and we have not studied this aspect. To avoid any misinterpretation of our goals, we change the title.

Round 2

Reviewer 3 Report

Comments and Suggestions for Authors

The authors have addressed my concerns in the revised manuscript. It is much clearer and the conclusions are better supported. I appreciate the additional effort on the part of the authors to make these revisions, and hope they agree that the paper is now stronger.

I have one additional suggestion, and that is to present the data that is currently on lines 189-199 into a chart, to improve readability.

Author Response

Thank you for the suggestion. According to the comment, we changed the data regarding the respondents' country of origin into a chart. To maintain a consistent representation of data throughout the manuscript, we have modified the data of both postgraduate and undergraduate students.